# Long-term maxillary anteroposterior changes following maxillary protraction with or without expansion: A meta-analysis and meta-regression

Wei-Cheng Lee[1,2], Yi-Shing Shieh[3], Yu-Fang Liao[2,4], Cho-Hao Lee[5], Chiung Shing Huang[2,4] *

1 Division of Orthodontics and Dentofacial Orthopedics, Department of Dentistry, Tri-Service General Hospital, National Defense Medical Center, Taipei, Taiwan, 2 Graduate Institute of Craniofacial and Dental Science, College of Medicine, Chang Gung University, Taoyuan, Taiwan, 3 Department of Dentistry, Tri-Service General Hospital, National Defense Medical Center, Taipei, Taiwan, 4 Faculty of Dentistry, Chang Gung Craniofacial Research Center, Chang Gung Memorial Hospital at Taipei, Chang Gung University College of Medicine, Taoyuan, Taiwan, 5 Division of Hematology and Oncology Medicine, Department of Internal Medicine, Tri-Service General Hospital, National Defense Medical Center, Taipei, Taiwan

* ortholab88@gmail.com

**Data Availability Statement:** All relevant data are within the manuscript and its Supporting Information files.

## Abstract

### Background

Maxillary protraction with or without expansion is a widely known orthopedic treatment modality in growing skeletal Class III patients. However, limited data are available regarding the outcomes of long-term changes in the maxilla. Aim of this meta-analysis was to assess the effectiveness of the long-term maxillary anteroposterior changes following a facemask therapy with or without rapid maxillary expansion in growing skeletal Class III patients.

### Methods

A comprehensive literature search was conducted using the databases of PubMed, Science Direct, Web of Science, and Embase. Randomized controlled trials and cohort studies, published up to Sep. 2020, with maxillary protraction and/or expansion as keywords were included in this meta-analysis. Risk of bias within and across studies were assessed using the Cochrane tools (RoB2.0 and ROBINS-I) and GRADE approach. Overall and subgroup comparisons with the random-effect model were performed in this meta-analysis. Meta-regression models were designed to determine potential heterogeneity.

### Results

There was a statistically significant increase (Mean difference, 2.29˚; 95% confidence interval, 1.86–2.73; and $p < 0.001$ after facemask (FM) protraction. Mean difference, 1.73˚; 95% confidence interval, 1.36–2.11; and $p < 0.001$ after rapid maxillary expansion(RME) and facemask protraction) in the Sella-Nasion-A point (SNA) angle in the treatment groups as compared with the control groups, when measured during the less than 3-year follow-up

**Funding:** The author(s) received no specific funding for this work.

**Competing interests:** The authors have declared that no competing interests exist.

period. However, no statistically significant changes (Mean difference, 0.28˚; 95% confidence interval, -0.57–1.13; and $p$ = 0.52 after facemask protraction. Mean difference, 0.34˚; 95% confidence interval, -0.64–1.33; and $p$ = 0.50 after rapid maxillary expansion and facemask protraction) were observed in the SNA angle in the groups, when measured after 3 years of follow-up. Meta-regression analysis also showed that with increased follow-up duration, the effectiveness of maxillary protraction decreased.

## Conclusion

This meta-analysis revealed that maxillary protraction therapy could be effective for a short-term in correcting maxillary hypoplasia and the treatment result was not affected by mean age and sex. However, with increased follow-up duration, the sagittal maxillary changes gradually decreased. Limitations on this review were only the SNA angle was used and clinical heterogeneity was not discussed. The quality of evidence was moderate. Further long-term observational studies are necessary for a comprehensive evaluation of the effects on maxillary skeletal changes.

## Introduction

The prevalence of skeletal class III malocclusion varies in different populations. Based on some studies, the prevalence of Class III malocclusion is approximately 1% to 5% in white populations and around 9% to 19% in Asian populations [1, 2]. In skeletal Class III malocclusion, the etiology is multifactorial including genetic inheritance, ethnic, environmental and habitual components [3] and genetic is the main etiology of skeletal Class III malocclusion [4]. According to surveys, 75% of skeletal Class III malocclusions are associated with maxillary retrognathism or a combination of maxillary retrognathism and mandibular prognathism [5]. In addition, nearly 30 to 40% of patients display some degree of maxillary deficiency [6]. Several studies also claimed that maxillary retrognathism is the most common contributing component of Class III characteristics [3, 7]. Thus, using maxillary protraction devices to enhance maxillary growth become more important [3, 7]. Furthermore, early treatment of growing patients with skeletal CIII malocclusion could provide them higher quality of life and make them more confident throughout the years they are most vulnerable by how they look like [8, 9]. Growing patients with skeletal Class III midfacial hypoplasia have been treated satisfactorily by orthopedic treatment of maxillary protraction with or without maxillary expansion [10–15]. In the past few years, facemask (FM) and rapid maxillary expansion (RME) were combined as a treatment modality for improving the maxillary transverse and midface deficiency. Another treatment option introduced was alternate rapid maxillary expansion and constriction, to open the circummaxillary sutures before maxillary protraction [16]. Furthermore, bone-anchored maxillary protraction is another recently developed method to enhance the therapeutic influence on midface deficiency [13, 17–20]. The correction of skeletal Class III malocclusion is challenging in orthodontics due to the unpredictable growth potential of the maxilla and potentially unfavorable mandibular growth.

Application of the FM protraction therapy in growing children with skeletal CIass III malocclusion is considered as a feasible treatment option for maxillary advancement [14, 15, 21]. The FM treatment has also been advised during the early orthopedic treatment of Class III malocclusion with maxillary deficiency [10, 22]. However, in the long-term observational

studies, the results were inconsistent [23, 24] and the skeletal effect on the reinforcement of maxillary growth over time from the traditional methods has been debated, and remains controversial. Statistically significant maxillary changes were observed after FM with or without RME treatment in some studies [13, 17, 21, 25, 26]. In contrast, limited or no significant evidence was observed in others [12, 14, 22, 27]. The major limitations among these studies were the lack of long-term follow-up [11, 13, 17, 23, 25, 28], absence of untreated control groups [29–31], and differences in the follow-up durations or treatment timing among studies [23, 32–34].

Even though several studies evaluating maxillary anteroposterior effects following maxillary protraction have been reported, most are conflicting results and still uncertain. Therefore, we systematically searched and analyzed the available literature for the advancement of scientific knowledge and clinical decision making. The purpose of this study was to evaluate the long-term maxillary anteroposterior changes following FM treatment with or without RME in growing skeletal Class III patients when compared to that in the untreated control group through meta-analysis and meta-regression.

## Materials and methods

Preferred Reporting Items for Systematic Reviews and Meta Analyses (PRISMA) [35] guidelines was adhered to perform systematic reviews and meta-analyses. This review protocol was also registered with the Open Science Framework platform (protocol available at osf.io/39kfs).

### Search strategy

Studies that described growing patients with skeletal Class III midfacial hypoplasia who received orthopedic treatment of maxillary protraction with or without expansion were included. Further, the skeletal changes after orthopedic treatment with FM or FM+RME were assessed and compared to that of the untreated control groups.

This meta-analysis aimed to determine whether any maxillary anteroposterior changes exist in those who need maxillary protraction with or without expansion. Four electronic databases, namely PubMed, Science Direct, Embase, and Web of Science, were searched to identify studies. This search included "maxilla constriction" or "midfacial deficiency" or "maxillary retrognathism" or "Class III malocclusion" AND "maxillary protraction" or "FM" or "facemask" or "reverse headgear" or "rapid maxillary expansion" or "maxillary expansion" or "RME" or"early treatment" or "orthopedic" AND "children/adolescence" or "growing" or "growth" AND "randomized controlled trial" or "randomized" or "randomly" or "RCT" or "cohort study" or "cohort" or "prospective" or "retrospective" or "controlled clinical trial". A detailed description of the search strategy applied to PubMed is provided in Table 1. In the extracted studies, references were evaluated to meet the following inclusion and exclusion criteria. Additionally, a manual search was carried out through the reference lists of the finally included articles, and the relevant systematic reviews and orthodontic journals not indexed in database.

**Table 1. Search strategy in Pubmed.**

| | |
|---|---|
| #1 | "maxilla constriction" or "midfacial deficiency" or "maxillary retrognathism" or "Class III malocclusion" |
| #2 | "maxillary protraction" or "FM" or "facemask" or "reverse headgear" or "rapid maxillary expansion" or "maxillary expansion" or "RME" or"early treatment" or "orthopedic" |
| #3 | "children/adolescence" or "growing" or "growth" |
| #4 | "randomized controlled trial" or "randomized" or "randomly" or "RCT" or "cohort study" or "cohort" or "prospective" or "retrospective" or "controlled clinical trial" |
| #5 | #1 AND #2 AND #3 AND #4 |

## Inclusion and exclusion criteria

The PRISMA checklist is described in the S1 Table. The included studies were cohort studies and randomized controlled trials (RCTs) with at least 6 months of follow-up that were published until September 2020 without language restrictions. Other inclusion criteria were following the PICOS principle. Type of participant (P), the patients selected were those with skeletal Class III malocclusion with maxillary hypoplasia or transverse maxillary deficiency, from the early mixed dentition to early permanent dentition (age ranged from 6 to 16 years). Type of interventions (I), the intervention was the selection of different treatment of FM and FM/RME. We performed two different types of comparisons (C) separately: 1) FM vs. control, 2) FM/RME vs. control in the long-term follow up. The outcome (O) of maxillary changes in sagittal dimensions, defined as Sella-Nasion-A point (SNA), was obtained by cephalometric radiography. Studies that satisfied the inclusion criteria were retrieved and screened using the following exclusion criteria: (1) patients with craniofacial anomalies, (2) No CIII malocclusion and (3) less than 6 months of follow-up.

## Data extraction

Among the included studies, we extracted and collected the following variables in a standardized form: authors, publication years, study design, patient classification, number of participants, mean age, sex, follow-up period, measurement method, and the clinical outcome. Three reviewers (WCL, YFL, and CHL) individually verified the data in the included studies. Subsequently, we overcame disagreements by means of discussion with the help of a fourth reviewer (CSH) to make the final decision.

## Risk of bias in individual studies

Four authors (WCL, YFL, CHL, and CSH) evaluated each RCT or controlled clinical trial's quality according to revised Cochrane risk of bias (RoB 2.0) [36] or risk of bias in non-randomized studies of interventions (ROBINS-I) [37], respectively. The quality assessments in the RoB 2.0 included the bias in the randomization process, deviations from the intended interventions, missing outcome data, measurement of the outcome, selection of the reported result, and overall bias. The quality assessments in the ROBINS-I included the bias in the pre-intervention, at intervention, post-intervention, and overall bias. In addition, the quality of the resultant evidence was assessed by the Grading of Recommendations Assessment, Development and Evaluation (GRADE) [38].

## Statistical analysis

We used the OpenMetaAnalyst software to obtain the mean difference (MD) and 95% confidence interval (CI). We used MD for continuous data in statistical pooling. We also used the $I^2$ statistical test to evaluate the heterogeneity of the included studies. An $I^2$ value ranged from 0 to 100%. An $I^2$ value = 0% meant there was no heterogeneity and $I^2$ value $\geq$ 50% proposed considerable heterogeneity [39]. We explored the source of heterogeneity by meta-regression using an average summary value. Possible moderators (age, sex, publication year, follow up period and study design) were tested to explore heterogeneity. And then we conducted a subgroup analysis from the meta-regression result. We used the OpenMetaAnalyst and Comprehensive Meta-Analysis software version 3 to perform meta-regression analysis, and subgroup analysis. Funnel plots were used to explore potential small study bias via visual inspection and Egger's test.

## Results

### Search results and description

**Characteristics of the included studies.** The PRISMA flow diagram is presented in Fig 1. Three hundred and twenty-nine articles were identified from the databases and other sources. Fifty-eight full-text articles were evaluated for eligibility and after 41 exclusions, 17 articles were included in this meta-analysis. The studies included were published between 1996 and 2016. Of the 17 included studies, four studies were RCTs and 13 studies were cohort studies. 10 studies [17, 22, 25–27, 40–44] were categorized into the FM group; whereas, eight studies [11–15, 21, 26, 28] were allotted to the FM+RME group. In the FM versus control group comparison, patients' ages ranged from 6.36 to 11.54 years and the follow-up period ranged between 6 months and 6 years. In the FM+RME versus control group comparison, patients' ages ranged between 6.4 and 10.91 years and the follow-up period ranged between 6.78 months and 9 years. The characteristics of the included studies are presented in Table 2.

### Assessment of risk of bias

Four of the included studies were RCTs and we evaluated the risk of bias using the RoB 2.0 tool. Four RCTs were found to have a low risk of bias. For observational studies, we used the ROBINS-I tool to classify the risk of bias among the studies into one of the four levels (low, moderate, serious, and critical). The overall result of the assessment showed that eight studies presented a low risk of bias, while the other five were at moderate risk of bias (Table 3). The most difficult domains involved were selection bias. The FM group included three RCTs and seven cohort studies that presented a moderate risk of bias in three cohort studies, while the others presented a low risk of bias. The FM+RME group included two RCTs and six cohort studies that presented a moderate risk of bias in three cohort studies, while the others presented a low risk of bias.

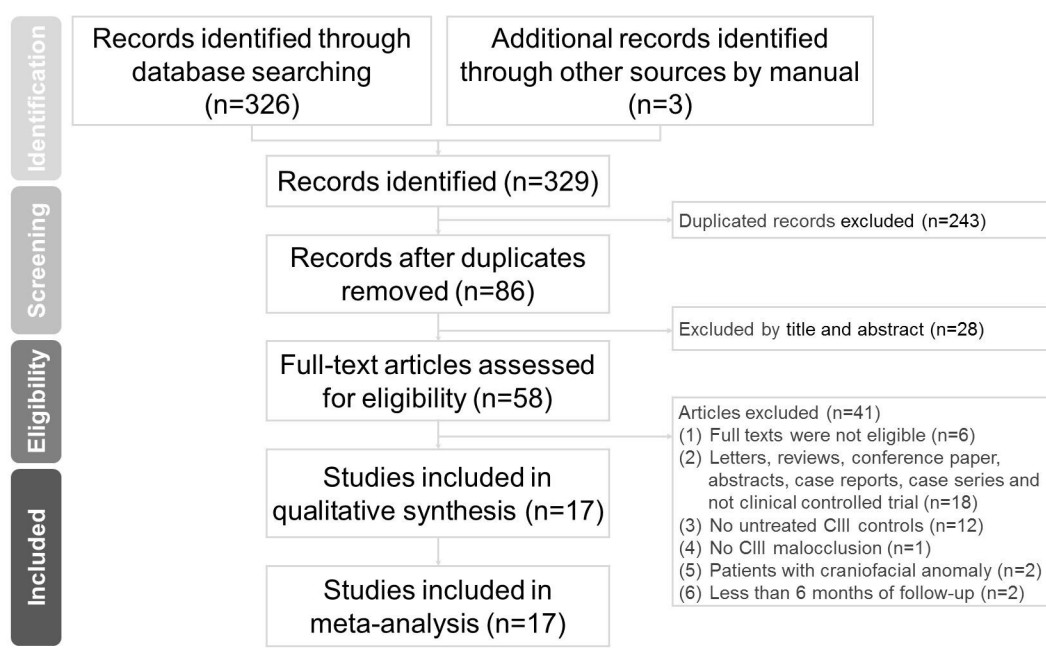

**Fig 1. PRISMA flow diagram of the search results from the databases.**

**Table 2. Characteristics of included studies (n = 17).**

| Author, year | Design | Type of malocclusion | Appliance (type of intervention) | Number | Mean age in years | Mean follow-up period | Outcomes |
|---|---|---|---|---|---|---|---|
| Chong, 1996 | CS (TG prospective) (CG retrospective) | Skeletal CIass III (negative OJ and/ or mesial step in postlactal plane.) | A = FM | n = 16 | 6.80 ± 1.13 | 3.57 years | Skeletal changes: SNA |
| | | | B = untreated control | n = 13 | 6.36 ± 0.54 | | |
| Kilicoglu, 1998 | CS (prospective) | Skeletal Class III, Angle Class III (ANB < -1˚) | A = FM | n = 16 (M = 0, F = 16) | 8.6 ± 1.4 | 12 months | Skeletal changes: SNA |
| | | | B = untreated control | n = 10 (M = 0, F = 10) | 9.2 ± 1.4 | | |
| Ucem, 2004 | CS (prospective) | Skeletal Class III (maxillary retrusion or a combination of maxillary retrusion and mandibular protrusion) | A = FM | n = 14 (M = 7, F = 7) | 10.4 | 9 months | Skeletal changes: SNA |
| | | | B = untreated control | n = 14 (M = 8, F = 6) | 9.67 | 11 months | |
| Vaughn. 2005 | RCT | Skeletal Class III, Angle Class III (ANB○< 0○; Nperp-A<2; Wits < -3;) | A = FM | n = 15 (M = 7, F = 8) | 7.3 | 1.16 year | Skeletal changes: SNA |
| | | | B = FM/RME | n = 14 (M = 7, F = 7) | 8.1 | 1.15 year | |
| | | | C = untreated control | n = 17 (M = 10, F = 7) | 6.6 | 1 year | |
| Cozza, 2010 | CS (TG prospective) (CG retrospective) | Skeletal Class III (Wits < -2, anterior crossbite or edge to edge, and CIII molar relationship) | A = FM | n = 22 | 8.9 | 2.1 years | Skeletal changes: SNA |
| | | | B = untreated control | n = 12 | 7.6 | | |
| Mandall, 2012 | RCT | Skeletal Class III (SNA, SNB, ANB) | A = FM | n = 35 | 8.7 | 3 years | Skeletal changes: SNA |
| | | | B = untreated control | n = 38 | 8.7 | | |
| Chen, 2012 | CS (prospective) | Skeletal Class III (ANB < 1 degree) | A = FM | n = 22 (M = 12, F = 10) | 11.38 ± 0.69 | 3 year | Skeletal changes: SNA |
| | | | B = untreated control | n = 17 (M = 7, F = 10) | 11.54 ± 1.07 | 1.75±0.83 year | |
| Akin, 2015 | CS (retrospective) | Skeletal Class III (ANB < 0˚, concave facial profile, anterior crossbite or edge to edge, CIII molar relationship | A = FM | n = 25 (M = 10, F = 15) | 10.3±1.5 | 6 months | Skeletal changes: SNA |
| | | | B = untreated control | n = 17 (M = 8, F = 9) | 10.1±1.3 | 6 months | |
| Baloş, 2015 | CS (retrospective) | Skeletal Class III | A = FM | n = 17 (M = 9, F = 8) | 11.3±1.0 | 1 year | Skeletal changes: SNA |
| | | skeletal (ANB < 0˚, SNA < 82˚) | B = untreated control | n = 11 (M = 8, F = 3) | 10.6±1.2 | 1 year | |
| Mandall, 2016 | RCT | Skeletal Class II (SNA, SNB, ANB) | A = FM | n = 35 | 8.7 ± 0.9 | 6 years | Skeletal changes: SNA |
| | | | B = untreated control | n = 32 | 9 ± 0.8 | 6 years | |
| Yuksel, 2001 | CS (prospective) | Skeletal and dental Class III malocclusion (reverse overjet and other cephalometric findings) | A = FM/RME | n = 17 (M = 11, F = 6) | 9.67 | 7 months | Skeletal changes: SNA |
| | | | B = untreated control | n = 17 (M = 11, F = 6) | 9.42 | 9 months | |

(*Continued*)

**Table 2.** (Continued)

| Author, year | Design | Type of malocclusion | Appliance (type of intervention) | Number | Mean age in years | Mean follow-up period | Outcomes |
|---|---|---|---|---|---|---|---|
| Xu, 2001 | RCT | Skeletal Class III (anterior crossbite and other cephalometric findings) | A = FM/RME | n = 20 | 9.3 | 11.3 months | Skeletal changes: SNA |
| | | | B = untreated control | n = 17 | 9.3 | 11.3 months | |
| Westwood, 2003 | CS (retrospective) | Skeletal Class III (Wits < -1.5, anterior crossbite or edge to edge) | A = FM/RME | n = 34 (M = 14, F = 20) | 8.25 ± 1.83 | 6.33 ± 2.25 years | Skeletal changes: SNA |
| | | | B = untreated control | n = 22 (M = 9, F = 13) | 8.08 ± 2.16 | 6.42 ± 2.17 years | |
| Kajiyama, 2004 | CS (retrospective) | Skeletal Class III (concave profiles, retrusive maxilla with or without mandibular protrusion, negative overjet, and other cephalometric findings indicating a Class III skeletal pattern) | A = FM/RME | n = 29 (M = 11, F = 18) | 8.58 ± 1.42 | 10.2± 4.5 months | Skeletal changes: SNA |
| | | | B = untreated control | n = 25 (M = 10, F = 15) | 8.83 ± 1.33 | 8.4 ± 2.3 months | |
| Masucci, 2011 | CS (prospective) | Skeletal Class III (Wits < -2, no CO CR discrepancy) | A = FM/RME | n = 22 (M = 9, F = 13) | 9.2±1.6 | 9.4±2.5 years | Skeletal changes: SNA |
| | | | B = untreated control | n = 13 (M = 8, F = 5) | 8.4±0.9 | 9.5±1.8 years | |
| Sar, 2011 | CS (prospective) | Skeletal Class III (ANB○< 0○; Nperp-A<1;Wits < -2;) | A = MP+FM | n = 15 (M = 10, F = 5) | 10.91± 1.22 | 6.78 months | Skeletal changes: SNA |
| | | | B = FM/RME | n = 15 (M = 8, F = 7) | 10.31± 1.52 | 9.45 months | |
| | | | C = untreated control | n = 15 (M = 7, F = 8) | 10.05± 1.14 | 7.59 months | |
| Masucci, 2014 | CS (prospective) | Skeletal Class III (Wits < -2, no CO CR discrepancy, anterior crossbite or edge-to-edge, mesial step relationships of the primary second molars or Class III relationships of the permanent first molars) | A = FM/Alt-RAMEC | n = 31 (M = 17, F = 14) | 6.4 ± 0.8 | 1.7 ± 0.4 years | Skeletal changes: SNA |
| | | | B = FM/RME | n = 31 (M = 16, F = 15) | 6.9 ± 1.1 | 1.6 ± 0.6 years | |
| | | | C = untreated control | n = 21 (M = 9, F = 12) | 6.5 ± 1.0 | 1.5 ± 0.4 years | |

RCT, randomized controlled trial; CS, cohort study; FM, facemask; RME, rapid maxillary expansion; Alt-RAMEC, alternate rapid maxillary expansion and constriction; SNA, Sella-Nasion-A point; TG, treated group; CG, untreated control group.

## Quantitative data synthesis

**Primary outcome on the SNA.** Primary outcomes on the SNA are shown in Fig 2. There were total 715 participants of the 17 studies included in the quantitative data synthesis as follows: 223 in the FM group, 182 in the FM+RME group, and 310 in the untreated control group. The results of the performed meta-analyses are given in Table 4. In the FM versus control group comparison, the pooled data demonstrated that the FM therapy had better treatment effect on the SNA (mean difference, 1.79˚; 95% CI, 1.20–2.39; and $I^2$ = 54.96%). However, significant heterogeneity was seen among the included studies. Similarly, in the FM

**Table 3. Methodological quality assessment of included studies.**

**Randomized controlled trials evaluated using the revised Cochrane risk of bias (RoB 2.0) tool.**

| Author, year | Bias arising from the randomization process | Bias due to deviations from the intended interventions | Bias due to missing outcome data | Bias in the measurement of the outcome | Bias in the selection of the reported result Low | Overall bias |
|---|---|---|---|---|---|---|
| Vaughn, 2005 | Low | Low | Low | Low | Low | Low |
| Mandall, 2012 | Low | Low | Low | Low | Low | Low |
| Mandall, 2016 | Low | Low | Low | Low | Low | Low |
| Xu, 2001 | Low | Low | Low | Low | Low | Low |

**Non-randomized controlled trial studies evaluated using the risk of bias in non-randomized studies of interventions (ROBINS-I) tool.**

| | Pre-intervention | | At intervention | Post-intervention | | | | Overall bias |
|---|---|---|---|---|---|---|---|---|
| Author, year | Bias due to confounding | Selection bias | Bias in the classification of interventions | Deviation from the intended interventions | Bias due to missing data | Bias in the measurement of outcomes | Bias in the selection of reported results | |
| Chong, 1996 | Low | Low | Low | Low | Low | Low | Low | Low |
| Kilicoglu, 1998 | Low | Low | Low | Low | Low | Low | Low | Low |
| Ucem, 2004 | Low | Low | Low | Low | Low | Low | Low | Low |
| Cozza, 2010 | Low | Low | Low | Low | Low | Low | Low | Low |
| Chen, 2012 | Low | Low | Low | Low | Low | Moderate | Low | Moderate |
| Akin, 2015 | Low | Moderate | Low | Low | Low | Low | Low | Moderate |
| Baloş, 2015 | Low | Moderate | Low | Low | Low | Low | Low | Moderate |
| Yuksel, 2001 | Low | Low | Low | Low | Low | Low | Low | Low |
| Westwood, 2003 | Low | Moderate | Low | Low | Low | Low | Low | Moderate |
| Kajiyama, 2004 | Low | Moderate | Low | Low | Low | Low | Low | Moderate |
| Masucci, 2011 | Low | Low | Low | Low | Low | Low | Low | Low |
| Sar, 2011 | Low | Low | Low | Low | Low | Low | Low | Low |
| Masucci, 2014 | Low | Low | Low | Low | Low | Low | Low | Low |

+RME versus control group comparison, the pooled data also demonstrated that the FM +RME therapy had better treatment effect on the SNA (mean difference, 1.54˚; 95% CI, 1.06–2.02; and $I^2$ = 41.59%). Significant heterogeneity was also seen among included studies.

## Meta-regression results

Table 5 shows the results of a meta-regression that investigated the origin of significant association ($p < 0.1$). All potential factors including mean age, sex, publication years, and study design did not present significant associations in this meta-analysis with the exception of follow-up period. Meta-regression model was developed to assess the amount of heterogeneity based on the study characteristics with respect to the SNA angle between treatment groups and untreated control groups. Regarding the difference between the SNA angle, a significant relationship was noted during the follow-up period in the FM or FM+RME groups in contrast to the untreated control group (Fig 3). Based on this meta-regression result, we conducted a subgroup analysis involving

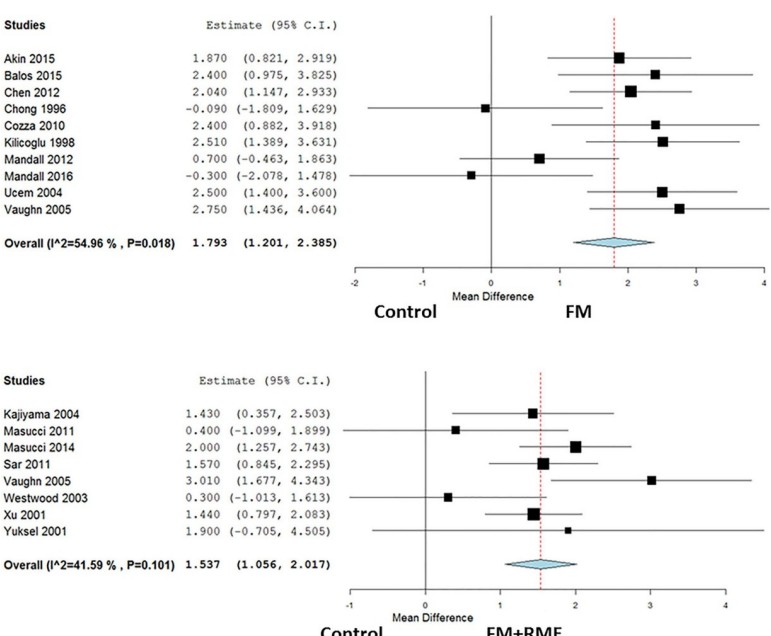

**Fig 2. Forest plots to evaluate maxillary anteroposterior changes in the SNA following maxillary protraction with or without expansion.** Fig 2(A). The FM treated group versus control group. Fig 2(B). The FM+RME treated group versus control group.

groups of participants with follow-up period less than 3 years and more than 3 years. This subgroup analysis demonstrated a significantly lower heterogeneity in each group.

From the meta-regression plot correction, we determined that follow-up period less than 3 years correlated with higher efficacy. However, the efficacy gradually reduced in the long-term follow-up period. The point of determination for difference in efficacy was approximately 3 years of follow-up.

## Subgroup analysis in the SNA

SNA changes from subgroup analysis of follow-up periods of less than and more than 3 years (Fig 4) were recorded and discussed. The results of the performed meta-analyses are given in Table 4.

*The FM treated group versus untreated control group.* The overall mean difference in the FM treated group versus the untreated control group regarding SNA angle was 1.79˚ (95% CI, 1.20–2.39 and $p < 0.001$ for the FM treated group). The subgroup analysis showed a

**Table 4. Summary results from primary and subgroup analyses.**

| Analysis | N | MD | 95% CI | *p* value | I² |
|---|---|---|---|---|---|
| **Primary outcome on SNA changes** | | | | | |
| FM versus untreated controls (follow up: range 6 months to 6 years) | 10 | 1.79 | 1.20 to 2.39 | $p<0.001$ | 54.96% |
| FM+RME versus untreated controls (follow up: range 6 months to 9 years) | 8 | 1.54 | 1.06 to 2.02 | $p<0.001$ | 41.59% |
| **Subgroup analysis on SNA changes** | | | | | |
| FM versus untreated controls (follow up: < 3 years) | 7 | 2.29 | 1.86 to 2.73 | $p<0.001$ | 0% |
| FM versus untreated controls (follow up: ≥ 3 years) | 3 | 0.28 | -0.57 to 1.13 | $p = 0.52$ | 0% |
| FM+RME versus untreated controls (follow up: < 3 years) | 6 | 1.73 | 1.36 to 2.11 | $p<0.001$ | 6.26% |
| FM+RME versus untreated controls (follow up: ≥ 3 years) | 2 | 0.34 | -0.64 to 1.33 | $p = 0.50$ | 0% |

**Table 5. Meta-regression analysis results.**

| Moderators | Variables | Study Number | *p*-value |
|---|---|---|---|
| **SNA changes via FM versus untreated group** | Mean age | 11 | 0.245 |
| | Sex | 7 | 0.164 |
| | Publication year | 11 | 0.360 |
| | Follow-up period | 11 | **0.001** |
| | Study design | 11 | 0.185 |
| **SNA changes via FM+RME versus untreated group** | Mean age | 8 | 0.358 |
| | Sex | 7 | 0.302 |
| | Publication year | 8 | 0.404 |
| | Follow-up period | 8 | **0.020** |
| | Study design | 9 | 0.962 |

SNA, Sella-Nasion-A point; FM, facemask; RME, rapid maxillary expansion.

significantly increased SNA angle with FM treatment than that in the untreated control group with a follow-up period of less than 3 years (Mean difference, 2.29˚; 95% confidence interval, 1.86–2.73; and p < 0.001 after facemask protraction), but not in the groups with more than 3 years of follow-up (Mean difference, 0.28˚; 95% confidence interval, -0.57–1.13; and *p* = 0.52 after facemask protraction). Regarding SNA angle heterogeneity, the $I^2$ was 54.96% in the overall included studies, less than 0.01% in the group with follow-up periods of less than 3 years, and less than 0.01% in the group with follow-up periods of more than 3 years.

*The FM+RME treated group versus untreated control group*. The overall mean difference in the FM+RME treated group versus the untreated control group regarding SNA angle was 1.54˚ (95% CI, 1.06–2.02 and p < 0.001 for the FM+RME treated group). The subgroup analysis showed a significantly increased SNA angle in the FM+RME treated group than in the untreated control group with follow-up period of less than 3 years (Mean difference, 1.73˚; 95% confidence interval, 1.36–2.11; and *p* < 0.001 after rapid maxillary expansion and facemask protraction), but not in the groups with follow-up period of more than 3 years (Mean difference, 0.34˚; 95% confidence interval, -0.64–1.33; and *p* = 0.5 after rapid maxillary expansion and facemask protraction). Regarding SNA heterogeneity, the $I^2$ was 41.59% in the overall included studies, 6.26% in the group with follow-up period of less than 3 years, and less than 0.01% in the group with follow-up period of more than 3 years.

**Publication bias.** Reporting biases are best performed only when we have a sufficient number in this study. And insufficient number of studies was included in this meta-analysis. Therefore, funnel plots were not performed in this meta-analysis.

**GRADE.** GRADE was used to assess overall evidence of both RCTs and observational studies in maxillary anteroposterior changes. Low quality of evidence shows that maxillary protraction may have benefit when compared to untreated control in SNA degree. The level of evidence for SNA changes was downgraded due to statistical heterogeneity and low number of included studies in outcome assessment. Summary of findings table according to GRADE approach was shown in Table 6.

## Discussion

### Summary of evidence

This meta-analysis assessed the long-term anteroposterior changes on the maxilla, defined as SNA, following maxillary protraction with or without expansion via different devices

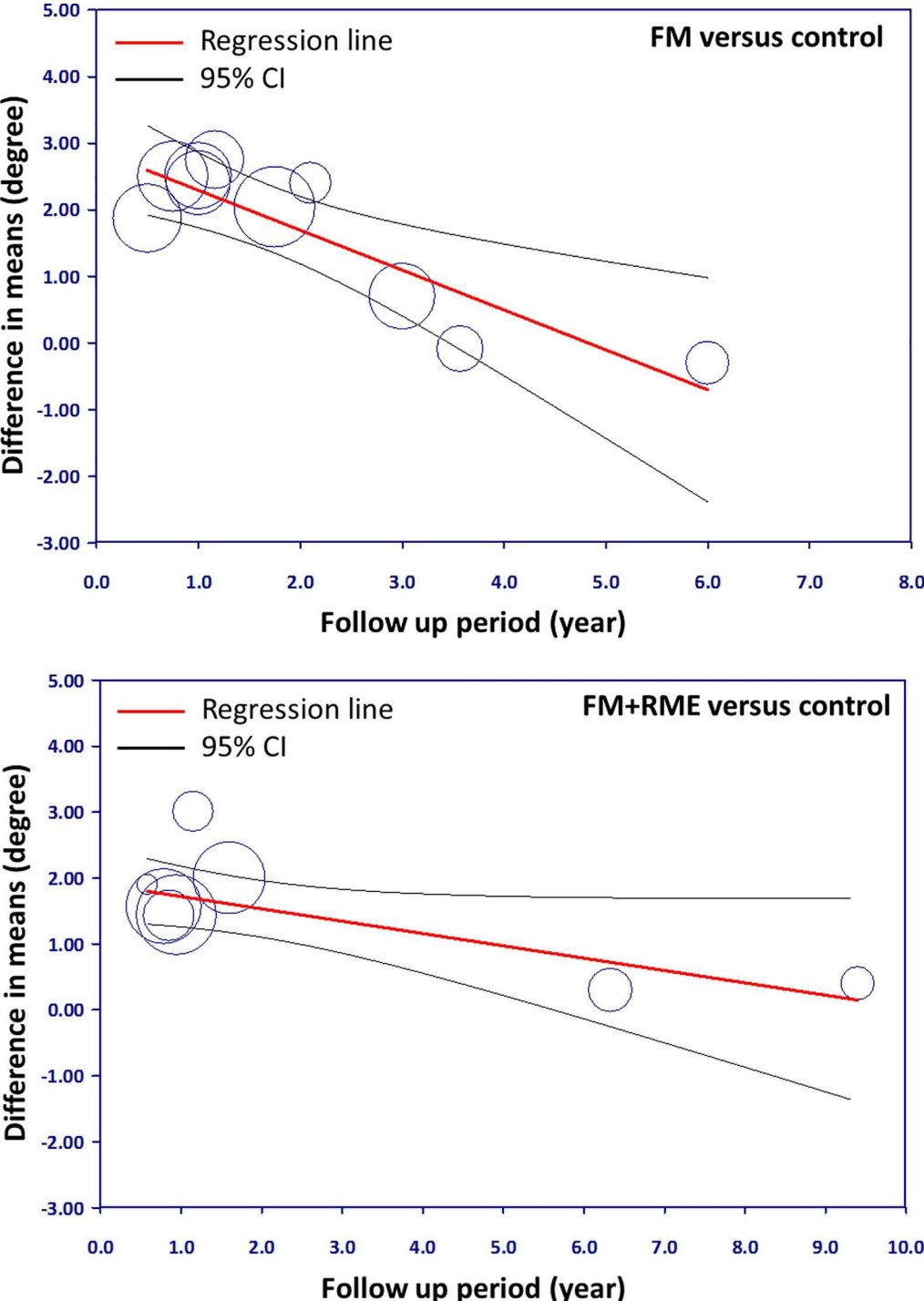

**Fig 3. Meta-regression plots of SNA changes and follow-up period.** Fig 3(A). The FM treated group versus control group. Fig 3(B). The FM+RME treated group versus control group.

including FM and FM+RME. This topic is not novel since many systematic reviews have been published in the past on similar topics [23, 45–49]. In the comparison between the FM treated group versus the untreated control group, 10 studies were included to investigate the orthopedic effects on the SNA. There was a significant increase in the SNA angle after FM treatment

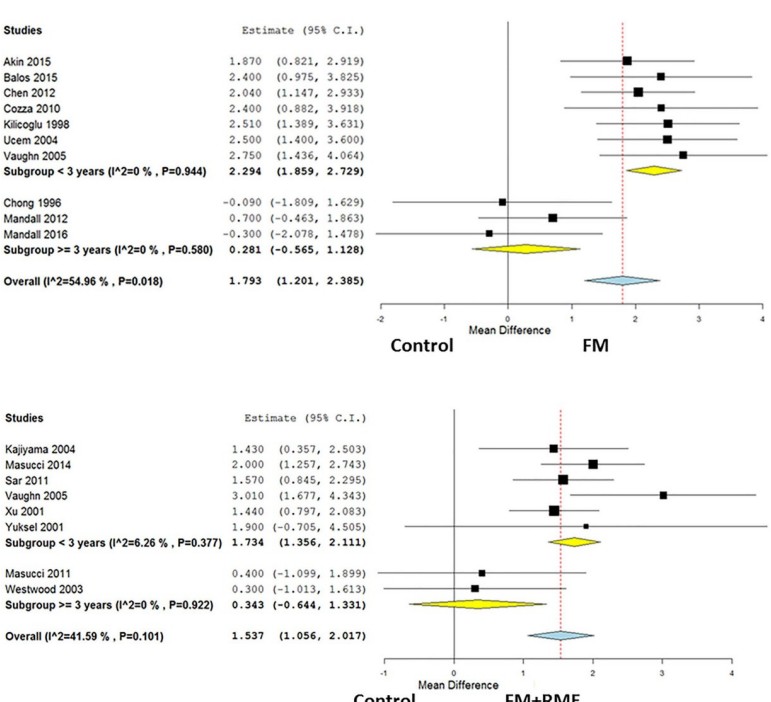

**Fig 4. SNA changes from subgroup analysis of follow-up periods of less and more than 3 years.** Fig 4(A). The FM treated group versus control group. Fig 4(B). The FM+RME treated group versus control group.

and it had similar effects on the SNA angle in the FM+RME treated group as compared with the untreated control group, which was consistent with the previous concept [5, 13, 17, 21]. Further, in the subgroup analysis of the FM treated group versus untreated control group, seven studies were included related to follow-up periods less than 3 years and three studies with follow-up periods more than 3 years. Patients undergoing FM treatment presented with a greater orthopedic effect on the SNA angle in the group with a follow-up period of less than 3 years when compared with the untreated control group. However, the effect was not significant in the group with more than 3 years follow-up period. Similarly, there was a greater orthopedic effect on the SNA angle after FM+RME treatment in the group with less than 3 years follow-up period. However, the effect was not significant in the group with more than 3 years follow-up period.

In this analysis, we included 17 studies (Table 2). Nevertheless, there was significant heterogeneity in the overall included studies in the FM or the FM+RME treatment group. The reason for this heterogeneity could be that the periods between the initial and final records were different among the included studies. Different follow-up durations of maxillary protraction may exist among studies, and this cannot be ignored when considering the potential origins of heterogeneity. Therefore, meta-regression models of the SNA angle differences were established with age, sex, follow-up period, and publication years as covariates (Table 5). In long-term follow-up periods, the effect on maxillary sagittal changes gradually decreased and became nearly equal to that in the control group with time [14, 22, 27]. Furthermore, other potential factors including mean age, sex, publication years, and study design could not significantly clarify heterogeneity in this meta-analysis.

Orthopedic maxillary protraction with or without expansion has been widely used for the treatment of the skeletal Class III growing patients with maxillary deficiency [10–15], and there have been several systematic reviews and meta-analyses [5, 23, 24, 45, 46, 49–51]

**Table 6. Overall summary of the evidence (GRADE).**

| № of studies | Study design | Risk of bias | Inconsistency | Indirectness | Imprecision | Other considerations | № of patients | | Effect | | Certainty | Importance |
|---|---|---|---|---|---|---|---|---|---|---|---|---|
| | | | | | | | Treated groups | untreated controls | Relative (95% CI) | Absolute (95% CI) | | |
| SNA changes (overall, FM versus untreated controls) (follow up: range 6 months to 6 years) | | | | | | | | | | | | |
| 3 | randomised trials | not serious | serious [a] | not serious | serious [b] | none | 77 | 82 | - | MD **1.11 degree higher** (0.58 lower to 2.8 higher) | ⊕⊕○○ LOW | IMPORTANT |
| SNA changes (overall, FM versus untreated controls) (follow up: range 6 months to 6 years) | | | | | | | | | | | | |
| 7 | observational studies | serious [c] | not serious | not serious | not serious | none | 148 | 107 | - | MD **2.07 degree higher** (1.55 higher to 2.58 higher) | ⊕⊕⊕○ MODERATE | IMPORTANT |
| SNA changes (overall, FM+RME versus untreated controls) (follow up: range 6 months to 9 years) | | | | | | | | | | | | |
| 2 | randomised trials | not serious | not serious | not serious | serious [b] | none | 35 | 34 | - | MD **2.11 degree higher** (0.59 higher to 3.63 higher) | ⊕⊕⊕○ MODERATE | IMPORTANT |
| SNA changes (overall, FM+RME versus untreated controls) (follow up: range 6 months to 9 years) | | | | | | | | | | | | |
| 6 | observational studies | serious [c] | not serious | not serious | not serious | none | 148 | 116 | - | MD **1.39 degree higher** (0.86 higher to 1.93 higher) | ⊕⊕⊕○ MODERATE | IMPORTANT |

CI: Confidence interval; **MD**: Mean difference

a. Downgraded one level for statistical heterogeneity

b. Downgraded one level for low number of included studies

c. Downgraded one levels for risk of bias within the included studies

investigating this treatment. A few studies [13, 17, 21, 25, 26] with orthopedic maxillary protraction reported a significant increase in the SNA angle. Other studies [24, 45–47, 50] found that protraction FM therapy in growing Class III patients is short-term effective. However, there was a lack of evidence on the long-term benefits, which remains controversial. Furthermore, conclusive evidence about the relationships between such changes and other potential factors, such as mean age, sex, publication years, and study design were lacking. In this analysis, our results showed that the patients who underwent maxillary protraction therapy (FM or FM+RME) with follow-up period of less than 3 years were likely to have an increased SNA angle than in the untreated control group. However, this benefit was not significant and maxillary anteroposterior changes gradually relapsed in the long-term follow-up period. In addition, the treatment timing was not affected by the early or late orthopedic treatment, which was similar to that reported in a previous study [5]. The treatment effect on maxillary anteroposterior changes was not affected by sex.

## Limitations and strengths

This study has several limitations. Firstly, only the SNA angle was used in this study as it was the most common denominator to represent the anteroposterior dimension of maxilla in various studies even though many other measurements were used [12, 14, 27]. Second, although we discussed the heterogeneity from the statistical point of view, we did not discuss clinical heterogeneity including the different treatment methods employed by different clinicians or the medical quality in the early periods, etc. The strength of this meta-analysis was that the studies we included were RCTs and observational studies instead of only RCTs. Admittedly, if the RCTs are blinded, they can supply the highest and reliable epidemiologic evidence for causality [52]. Observational studies were enrolled in this study, these studies may have strong probability of confounding and bias, are likely to have incomplete and poor quality of data, and less likely to have verifiable outcomes [53, 54]. Nevertheless, in particular conditions, observational studies may be of certain advantages. For example, they can provide us long-term investigation on orthopedic treatment of Class III malocclusion. Furthermore, in ethical issues, with patients that are seeking the treatment due to their orthopedic problems, observational studies may be more appropriate than RCTs in real-world circumstances as a result of the possibility of larger sample sizes, extensive participants included, and longer follow-up [52, 55]. However, in this analysis, few RCTs base were available. Instead, the included studies went through quality assessment (Table 3), meta-regression (Table 5 and Fig 3), and subgroup analyses (Fig 4) to evaluate the quality of evidence and heterogeneity.

This study investigated the relationship between maxillary anteroposterior changes following FM with or without RME. Certainly, some studies reported that maxillary protraction is significantly associated with the changes in the maxillary anteroposterior dimension, while other studies reported otherwise. This inconsistency was due to the different follow-up period in different included studies, and untreated control groups were not included in most studies. Furthermore, only the difference between initial and final records was compared between identically treated groups. Nevertheless, the maxillary changes were also associated with the effect of growth in children, and we included the untreated control group to decide the real effect of orthopedic maxillary protraction. Hence, we excluded case series studies resulting in reduced final sample size. Moreover, most studies evaluated the short-term effect, and did not include information regarding the population under study, age, sex, follow-up period, among others to investigate how these factors affected the treatment. Hence, we included studies from short-term to long-term follow-up period and conducted meta-regression analyses to evaluate heterogeneity in the included studies.

## Conclusion

Maxillary protraction treatments could be effective for a short-term in correcting maxillary hypoplasia in young patients and the treatment result was not affected by mean age and sex. Nevertheless, the skeletal effects gradually decreased with time in the long-term follow-up of maxillary sagittal changes. Hence, more high-quality long-term RCTs and observational studies are required to further evaluate the effects on maxillary skeletal changes.

## Supporting information

**S1 Table. PRISMA 2009 checklist.**
(DOC)

**S2 Table. List of included and excluded studies, with the corresponding reasons.**
(DOCX)

## Acknowledgments

The authors gratefully acknowledge the Center for Evidence-based Medicine, Tri-Service General Hospital, Taipei, Taiwan and Chang Gung Craniofacial Research Center, Taoyuan, Taiwan.

## Author Contributions

**Conceptualization:** Wei-Cheng Lee, Yu-Fang Liao, Cho-Hao Lee, Chiung Shing Huang.

**Data curation:** Wei-Cheng Lee, Cho-Hao Lee.

**Formal analysis:** Wei-Cheng Lee.

**Investigation:** Wei-Cheng Lee, Yu-Fang Liao, Chiung Shing Huang.

**Methodology:** Wei-Cheng Lee, Yu-Fang Liao, Cho-Hao Lee, Chiung Shing Huang.

**Resources:** Wei-Cheng Lee.

**Software:** Wei-Cheng Lee, Cho-Hao Lee.

**Supervision:** Yi-Shing Shieh, Chiung Shing Huang.

**Validation:** Yi-Shing Shieh.

**Writing – original draft:** Wei-Cheng Lee.

**Writing – review & editing:** Wei-Cheng Lee, Yu-Fang Liao, Cho-Hao Lee, Chiung Shing Huang.

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
