## [Decision Letter · Decision Letter 0]

22 Jul 2020

PONE-D-20-11899

Long-term maxillary anteroposterior changes following maxillary protraction with or without expansion: a meta-analysis and meta-regression

PLOS ONE

Dear Dr. Huang,

Thank you for submitting your manuscript to PLOS ONE. After careful consideration, we feel that it has merit but does not fully meet PLOS ONE’s publication criteria as it currently stands. Therefore, we invite you to submit a revised version of the manuscript that addresses the points raised during the review process.

Although the manuscript presents potential for publication in PLOS ONE, some major points raised by the reviewers must be addressed. We are looking forward to your revised manuscript.

We look forward to receiving your revised manuscript.

Kind regards,

Claudia Trindade Mattos, Ph.D.

Academic Editor

PLOS ONE

Journal Requirements:

2. Please provide the full electronic search strategy for at least one database, including any limits used, such that it could be repeated.

Reviewers' comments:

Reviewer's Responses to Questions

**Comments to the Author**

1. Is the manuscript technically sound, and do the data support the conclusions?

Reviewer #1: Partly

Reviewer #2: Partly

2. Has the statistical analysis been performed appropriately and rigorously? 

Reviewer #1: No

Reviewer #2: No

3. Have the authors made all data underlying the findings in their manuscript fully available?

Reviewer #1: No

Reviewer #2: Yes

4. Is the manuscript presented in an intelligible fashion and written in standard English?

Reviewer #1: Yes

Reviewer #2: Yes

5. Review Comments to the Author

Reviewer #1: The article addresses a relevant topic for orthodontics. However, revisions needed to be made to improve the article and strengthen the study's findings.

1) INTRODUCTION:

• Authors should cite more data showing the prevalence of skeletal Class III midfacial hypoplasia found at the population level.

• It is important to show data from studies that explain the etiology, clinical, epidemiological characteristics, and forms of treatment, highlighting their advantages, disadvantages and possible long-term effects.

• The impacts on quality of life can also be highlighted.

• Before the objectives, the authors should write the importance of studies like this for the advancement of scientific knowledge and clinical decision making.

• Is the review unprecedented? In the literature, I identified the existence of other systematic reviews on the topic. Therefore, the authors need to explain the differential of this review in relation to the others that have already been published and what it adds.

2) MATHERIAL AND METHODS:

• Was the review protocol previously registered in an online database, such as PROSPERO?

• In the methodology, it is recommended to insert a table with the search strategy used in each specific database, for reasons of search transparency and also because each database requires adaptations in the strategy.

• How was the selection and exclusion of duplicate references made? Manually? Using any software? The authors need to provide more details in the materials and methods on how the studies were selected, according to the PRISMA items.

• How was the examiner calibrated? Did the authors carry out any previous training? Was the kappa test to assess agreement between examiners calculated?

• The list of references for eligible studies needs to be consulted to identify possible articles that were not identified through searches.

• As it involves a review of clinical trials, it is recommended to apply the GRADE tool (Grading of Recommendations, Assessment, Development and Evaluations) to summarize the quality of available scientific evidence.

• Why did the authors not use the standardized mean difference in the meta-analysis instead of the mean difference? This detail needs to be justified.

3) RESULTS

• The results need to be described in more detail.

• In the abstract, the authors declare that: “However, no statistically significant changes (mean difference, 1.54°; 95% confidence interval, 1.06-2.02; and p < 0.001) were observed in the SNA angle in the groups, when measured after 3 years of follow-up.” If the p-value was significant, why was there no statistical difference? The results are conflicting.

4) DISCUSSION

• The discussion can be improved. The results of the studies need to be discussed in greater depth. The review included a large number of studies, but in the discussion, it is necessary to make clear the implications for the clinical practice of the results obtained. What are the strengths of this review compared to those already published on the topic? What can be better evaluated in future studies?

Reviewer #2: Summary

This is a potentially interesting study, but has several issues with the methods and reporting of it. These need to be taken into account and the study revised, before this can be further assessed for appropriateness. Thanks for letting me see this.

Specific comments

1. The abstract seems well-written. I would suggest however to also add the quality of evidence and existing limitations.

2. The last search date (Dec 19) is half a year old, therefore the review might be outdated.

3. Please provide the full exact search strategy for at least one database.

4. Why was January 1990 chosen as a date?

5. “Another inclusion criterion was adherence to the PICOS principle.”: What is meant with this?

6. “…age ranged from 6 to 16 years…”: why was this chosen? Was this criterion checked for all included studies?

7. “the intervention was the selection of different treatment of FM and FM/RME”: this is unclear, please explain.

8. “The outcome (O) of interest was long-term (lasting 9 years) maxillary changes in sagittal dimensions, defined as Sella-Nasion-A point (SNA).”: you should be here explaining the a priori methods. Not the results of your study.

9. The term ‘quality’ that you use when evaluating RCTs is not correct. Please check the PRISMA statement for the appropriate terminology.

10. Choosing between a fixed- or random-effects model according to the observed (calculated) heterogeneity is a very problematic method. The two models are not interchangeable and have different assumptions/method/interpretation. Please check with your statistician.

11. “…to perform sensitivity test, meta-regression analysis, and subgroup analysis.”: please explain in detail.

12. Reporting biases are best performed only when you have a sufficient number of studies at your disposal. Please check the recommendations for such tests.

13. The authors should provide a list with all included/excluded studies from their search together with the exclusion criteria.

14. Is it prudent to naively include/combine both RCTs and non-RCTs, since the latter can introduce additional bias in your results?

15. What kind of cohort studies were included? Prospective or retrospective? The latter are usually more biased than the former.

16. Please provide additional details in Table I regarding the patients’ (how was skel. Class III defined), the treatment (appliance, duration, etc), and the outcomes assessed. Also relaying the actual conclusions of each study might be confusing, since they might disagree with the results of the meta-analysis.

17. I am having issues with the risk of bias assessment presented. I checked randomly 2 included studies and I do not agree with the lenient assessment done with the authors. This needs to be checked to see if methods were appropriately applied.

18. The authors need to provide in a table the full results of all meta-analyses performed, including studies, estimate with imprecision, p value, and heterogeneity statistics.

19. How come the ‘explained variance’ is presented in table 3 but has not been described in the methods before? This is inappropriate. Also, reporting such a thing in meta-regressions that are not statistically significant is misleading. Please check with your statistician.

20. I see no mention of assessing the quality of evidence with the GRADE framework, even though this is a standard approach for all systematic reviews nowadays.

21. I see no clear section outlining the limitation of the present study.

22. Figure 2 is not necessary—you can give this information plainly in text.

6. PLOS authors have the option to publish the peer review history of their article (what does this mean?). If published, this will include your full peer review and any attached files.

Reviewer #1: No

Reviewer #2: **Yes: **Spyridon N. Papageorgiou

---

## [Author Response · Author response to Decision Letter 0]

27 Nov 2020

Dear editor and reviewers: 

We are extremely grateful to you for the constructive critique of our manuscript. We have responded to each of the comments of the referees on separate sheets and deeply appreciated your suggestions that have led to a significant improvement in this article. In response to your comments, we have revised the manuscript, tables and figures to enhance article readability. Several new sections of text are added. We have also reedited the abstract, results, methods, and discussion sections. All the changes are labeled in red color. Thanks for your encouragement and appreciation.

Reviewer(s)' Comments to Author: 

Recommendations for improvement: 

Reviewer #1: The article addresses a relevant topic for orthodontics. However, revisions needed to be made to improve the article and strengthen the study's findings.

1) INTRODUCTION:

• Authors should cite more data showing the prevalence of skeletal Class III midfacial hypoplasia found at the population level.

Remedy:

Yes, thank you for your suggestions. We added more data in the following statement: The prevalence of skeletal class III malocclusion varies in different populations. Based on some studies, the prevalence of Class III malocclusion is approximately 1% to 5% in white populations and around 9% to 19% in Asian populations.[1,2] (Please see Line 66, Page 4). According to surveys, 75% of skeletal Class III malocclusions are associated with maxillary retrognathism or a combination of maxillary retrognathism and mandibular prognathism [3]. In addition, nearly 30–40% of patients display some degree of maxillary deficiency [4] (Please see Line 71-74, Page 4)

• It is important to show data from studies that explain the etiology, clinical, epidemiological characteristics, and forms of treatment, highlighting their advantages, disadvantages and possible long-term effects.

Remedy:

Yes, thank you for your suggestions. We added more data in the following statement: 

In skeletal Class III malocclusion, the etiology is multifactorial including genetic inheritance, ethnic, environmental and habitual components[5] and genetic is the main etiology of skeletal Class III malocclusion [6]. According to surveys, 75% of skeletal Class III malocclusions are associated with maxillary retrognathism or a combination of maxillary retrognathism and mandibular prognathism [3]. In addition, nearly 30 to 40% of patients display some degree of maxillary deficiency [4]. Several studies also claimed that maxillary retrognathism is the most common contributing component of Class III characteristics.[5,7] Thus, using maxillary protraction devices to enhance maxillary growth become more important [5,7]. (Please see Line 66-75, Page 4). However, the correction of skeletal Class III malocclusion is challenging in orthodontics due to the unpredictable growth potential of the maxilla and potentially unfavorable mandibular growth. Even though several studies evaluating maxillary anteroposterior effects following maxillary protraction have been reported, most are conflicting results and still uncertain in the long-term following up. 

• The impacts on quality of life can also be highlighted.

Remedy:

Yes, thank you for your suggestions. The impact on quality of life was mentioned as following statement: Early treatment of growing patients with skeletal CIII malocclusion could provide them higher quality of life and make them more confident throughout the years they are most vulnerable by how they look like [8,9] (Please see Line 75-78, Page 4)

• Before the objectives, the authors should write the importance of studies like this for the advancement of scientific knowledge and clinical decision making.

Remedy:

Yes, thank you for your valuable suggestions. We added the importance of the study before the objectives as following: Even though several studies evaluating maxillary anteroposterior effects following maxillary protraction have been reported, most are conflicting results and still uncertain. Therefore, we systematically searched and analyzed the available literature for the advancement of scientific knowledge and clinical decision making. (Please see Line 105-108, Page 5)

• Is the review unprecedented? In the literature, I identified the existence of other systematic reviews on the topic. Therefore, the authors need to explain the differential of this review in relation to the others that have already been published and what it adds.

Remedy:

Yes, thank you for your suggestions. It is true that this topic is not novel since many systematic reviews have been published in the past on similar topics [3,10-15]. However, the follow up periods in most studies were short-term rather than long-term follow up. For clinicians, they hope the orthopedic treatment effect on growing skeletal CIII patients could be maintained to prevent the orthognathic surgery in the end of growth period. In addition, high heterogeneity was found in most systematic review and meta-analysis. In this article, we explored the heterogeneity using meta-regression and investigate that follow up periods was the main factor to cause the high heterogeneity. 

2) MATHERIAL AND METHODS:

• Was the review protocol previously registered in an online database, such as PROSPERO?

Remedy:

Yes, thank you for your valuable reminding. We registered this review protocol with the Open Science Framework platform (protocol available at osf.io/39kfs). (Please see Line 114-117, Page 6) 

• In the methodology, it is recommended to insert a table with the search strategy used in each specific database, for reasons of search transparency and also because each database requires adaptations in the strategy.

Remedy: 

Yes, thank you for your valuable suggestion. We insert a table with the search strategy in table 1 for reasons of search transparency. 

• How was the selection and exclusion of duplicate references made? Manually? Using any software? The authors need to provide more details in the materials and methods on how the studies were selected, according to the PRISMA items.

Remedy:

Yes, thank you for your suggestions. We selected the references into Endnote software and excluded the duplicated references. In addition, protocol section was added in the materials and methods (Please see Line 114-118, Page 6 ) and the studies was selected following the PRISMA items.(Please see S1 Table).

• How was the examiner calibrated? Did the authors carry out any previous training? Was the kappa test to assess agreement between examiners calculated?

Remedy:

Yes, thank you for your valuable suggestions. In order to measure inter-rater reliability, we measure the kappa test to assess agreement between examiners calculated. The kappa score was 0.85, indicating a high level of agreement.

• The list of references for eligible studies needs to be consulted to identify possible articles that were not identified through searches.

Remedy:

Yes, thank you for your suggestions. We re-organized the eligible studies from the databases and manual searching was done to find studies that may not have been indexed in the databases. In addition, we verified and discussed the eligible studies with other authors. 

• As it involves a review of clinical trials, it is recommended to apply the GRADE tool (Grading of Recommendations, Assessment, Development and Evaluations) to summarize the quality of available scientific evidence.

Remedy:

Yes, thank you for your valuable suggestions. We conducted the GRADE tool to summarize the quality of available scientific evidence.(Please see Line 356-364, Page 22 and table 6)

• Why did the authors not use the standardized mean difference in the meta-analysis instead of the mean difference? This detail needs to be justified.

Remedy:

Yes, thank you for your suggestions. All study effect sizes are weighted in a meta-analysis regardless to whether they are standardized mean differences, mean differences, odds ratios, risk ratios, correlations, etc. The mean difference is preferred when all studies use the same outcome (a continuous one) and unit of measure. For example, studies on the effect of a drug on blood pressure will almost all use mmHg as the unit of measure and if they don't, the reported value can be easily converted to mmHg. On the other hand, the standardized mean difference is used when the studies don't use the exact same outcome measure. For example, all studies measure depression but they use different psychometric scales Here we have to use the standardized mean difference. In this meta-analysis, all included studies use the same outcome (SNA and continues one) and unit of measure (degree). That is the reason why this meta-analysis uses the mean difference rather than standardized mean difference. 

Reference: https://training.cochrane.org/handbook/current

The standardized mean difference is used as a summary statistic in meta-analysis when the studies all assess the same outcome but measure it in a variety of ways (for example, all studies measure depression but they use different psychometric scales). In this circumstance it is necessary to standardize the results of the studies to a uniform scale before they can be combined. 

The mean difference (more correctly, ‘difference in means’) is a standard statistic that measures the absolute difference between the mean value in two groups in a clinical trial. It estimates the amount by which the experimental intervention changes the outcome on average compared with the control. It can be used as a summary statistic in meta-analysis when outcome measurements in all studies are made on the same scale.

3) RESULTS

• The results need to be described in more detail.

Remedy:

Yes, thank you for your suggestions. We described the results in more detail as following statement: 

The FM treated group versus untreated control group

The overall mean difference in the FM treated group versus the untreated control group regarding SNA angle was 1.79° (95% CI, 1.20-2.39 and p < 0.001 for the FM treated group). The subgroup analysis showed a significantly increased SNA angle with FM treatment than that in the untreated control group with a follow-up period of less than 3 years (Mean difference, 2.29°; 95% confidence interval, 1.86-2.73; and p < 0.001 after facemask protraction), but not in the groups with more than 3 years of follow-up (Mean difference, 0.28°; 95% confidence interval, -057-1.13; and p= 0.52 after facemask protraction). Regarding SNA angle heterogeneity, the I2 was 54.96% in the overall included studies, less than 0.01% in the group with follow-up periods of less than 3 years, and less than 0.01% in the group with follow-up periods of more than 3 years.

The FM+RME treated group versus untreated control group 

The overall mean difference in the FM+RME treated group versus the untreated control group regarding SNA angle was 1.54° (95% CI, 1.06-2.02 and p < 0.001 for the FM+RME treated group). The subgroup analysis showed a significantly increased SNA angle in the FM+RME treated group than in the untreated control group with follow-up period of less than 3 years (Mean difference, 1.73°; 95% confidence interval, 1.36-2.11; and p < 0.001 after rapid maxillary expansion and facemask protraction), but not in the groups with follow-up period of more than 3 years (Mean difference, 0.34°; 95% confidence interval, -0.64-1.33; and p= 0.5 after rapid maxillary expansion and facemask protraction). Regarding SNA heterogeneity, the I2 was 41.59% in the overall included studies, 6.26% in the group with follow-up period of less than 3 years, and less than 0.01% in the group with follow-up period of more than 3 years. (Please see Line 314-338 , Page 21-22 )

• In the abstract, the authors declare that: “However, no statistically significant changes (mean difference, 1.54°; 95% confidence interval, 1.06-2.02; and p < 0.001) were observed in the SNA angle in the groups, when measured after 3 years of follow-up.” If the p-value was significant, why was there no statistical difference? The results are conflicting.

Remedy:

Yes, thank you for your valuable suggestions and reminding. We corrected the paragraph as following: There was a statistically significant increase (Mean difference, 2.29°; 95% confidence interval, 1.86-2.73; and p < 0.001 after facemask (FM) protraction. Mean difference, 1.73°; 95% confidence interval, 1.36-2.11; and p < 0.001 after rapid maxillary expansion(RME) and facemask protraction) in the Sella-Nasion-A point (SNA) angle in the treatment groups as compared with the control groups, when measured during the less than 3-year follow-up period. However, no statistically significant changes (Mean difference, 0.28°; 95% confidence interval, -057-1.13; and p= 0.52 after facemask protraction. Mean difference, 0.34°; 95% confidence interval, -0.64-1.33; and p= 0.5 after rapid maxillary expansion and facemask protraction) were observed in the SNA angle in the groups, when measured after 3 years of follow-up. (Please see Line 40-50 , Page 2 )

4) DISCUSSION

• The discussion can be improved. The results of the studies need to be discussed in greater depth. The review included a large number of studies, but in the discussion, it is necessary to make clear the implications for the clinical practice of the results obtained. What are the strengths of this review compared to those already published on the topic? What can be better evaluated in future studies?

Remedy:

Yes, thank you for your suggestions. The implication for the clinical practice is the difference of orthodontic treatment time. From this review, it is shown that early treatment of growing patients with skeletal CIII malocclusion could provide them higher quality of life and make them more confident throughout the years they are most vulnerable by how they look like [8,9]. (Please see Line 75-78, Page 4). However, the patients may wear the orthodontic appliance (orthopedic + orthodontic treatment) for a long time (from young age to adult) and patient’s compliance may gradually decrease in the orthodontic treatment period. The strength of this review was that follow up periods were long-term rather short-term. In addition, the studies we included were RCTs and observational studies instead of only RCTs. (Please see Line 431, Page 26). About what can be better evaluated in future studies, we hope to evaluate long-term three–dimensional changes in different structures (ex: maxilla and mandible) after maxillary protraction. 

 

Reviewer #2:

This is a potentially interesting study, but has several issues with the methods and reporting of it. These need to be taken into account and the study revised, before this can be further assessed for appropriateness. Thanks for letting me see this.

Specific comments

1. The abstract seems well-written. I would suggest however to also add the quality 

of evidence and existing limitations.

Remedy:

Yes, thank you for your positive and encouraging comments on this manuscript. We add the quality of evidence and existing limitations as the following statement: Limitations on this review were only the SNA angle was used and clinical heterogeneity was not discussed. The quality of evidence was moderate. (Please see Line 57-59 , Page 3 )

2. The last search date (Dec 19) is half a year old, therefore the review might be outdated.

Remedy:

Yes, thank you for your suggestions. We followed your comments and re-searched the database up to Sep. 2020. (Please see Line 34, Page 2)

3. Please provide the full exact search strategy for at least one database.

Remedy:

Yes, thank you for your valuable suggestion. We insert a table with the search strategy in table 1 for reasons of search transparency.

4. Why was January 1990 chosen as a date?

Remedy:

Yes, thank you for your valuable suggestions and correction. We should not choose January 1990 as a date. Hence, we re-searched the database and corrected as the following statement: The included studies were cohort studies and randomized controlled trials (RCTs) with at least 6 months of follow-up that were published until September 2020. (Please see Line 157-159 , Page 8)

5. “Another inclusion criterion was adherence to the PICOS principle.”: What is meant with this?

Remedy:

Yes, thank you for your suggestions and correction. We corrected this sentence to “other inclusion criteria were following the PICOS principle.” (Please see Line 160-161, Page 8)

6. “…age ranged from 6 to 16 years…”: why was this chosen? Was this criterion checked for all included studies?

Remedy:

Yes, thank you for your valuable suggestions and alert. The treatment timing in growing patients with Class III malocclusion was ranged from the early mixed dentition to early permanent dentition[3]. The main objective of early facemask treatment is to enhance forward displacement of the maxilla by sutural growth. Histologic studies have shown that the midpalatal suture is broad and smooth during the“infantile” stage (8 to 10 years of age). The suture becomes more squamous and overlapping in the “juvenile” stage (10 to 13 years), and becomes more heavily interdigitated around puberty[16]. Some reports suggest that the optimal time for FM or FM / RME is before the age of 8 years [17-19]. It was also claimed that the suggested time of treatment in early orthopedic treatment is between the ages of 6 and 8 years after the maxillary permanent first molars and incisors have erupted[17]. However, other studies have found that patient’s age had little influence on treatment response and outcome [3,20]. The age of the participant was also divided into three time points: 6–8, 10–12, and 14–16 years of age to investigate craniofacial facial growth of CIII subjects [21]. From previous studies, the treatment timing for Orthopedic CIII treatment was inconsistent in different studies. Hence, we chose the age ranged from 6 to 16 years. We also checked all included studies and they were all in this range. 

7. “the intervention was the selection of different treatment of FM and FM/RME”: this is unclear, please explain.

Remedy:

Yes, thank you for your suggestions. The treatment of FM means facemask protraction therapy and FM/RME means the combination of facemask and rapid maxillary expansion.

8. “The outcome (O) of interest was long-term (lasting 9 years) maxillary changes in sagittal dimensions, defined as Sella-Nasion-A point (SNA).”: you should be here explaining the a priori methods. Not the results of your study.

Remedy:

Yes, thank you for your valuable suggestions. We changed this sentence as the following statement: We performed two different types of comparisons (C) separately: 1) FM vs. control, 2) FM/RME vs. control in the long-term follow up. The outcome (O) of maxillary changes in sagittal dimensions, defined as Sella-Nasion-A point (SNA), was obtained by cephalometric radiography. (Please see Line 166-171 , Page 8)

9. The term ‘quality’ that you use when evaluating RCTs is not correct. Please check the PRISMA statement for the appropriate terminology.

Remedy:

Yes, thank you for your valuable suggestions. We corrected the term “Quality assessments of the included studies” to “Risk of Bias in Individual Studies” when evaluating the included studies. (Please see Line 185, Page 8) 

10. Choosing between a fixed- or random-effects model according to the observed (calculated) heterogeneity is a very problematic method. The two models are not interchangeable and have different assumptions/method/interpretation. Please check with your statistician.

Remedy:

Yes, thank you for your valuable suggestions and reminding. It is true that it is a very problematic method by choosing between fixed or random effects models according to the observed (calculated) heterogeneity. In this meta-analysis, we check our statistician and explored the source of heterogeneity by meta-regression using an average summary value. Possible moderators (age, sex, publication year, follow-up period and study design) were tested to explore heterogeneity. And then we conducted a subgroup analysis from the meta-regression result. (Please see Line 202-206, Page 9)

11. “…to perform sensitivity test, meta-regression analysis, and subgroup analysis.”: please explain in detail.

Remedy:

Yes, thank you for your valuable suggestions. There are three major sources of heterogeneity: clinical, statistical and methodological heterogeneity. In order to explore the source of heterogeneity (I2 value >50% indicated a moderate

to high heterogeneity[22]), we conducted with meta-regression using an average summary value. Possible moderators (age, sex, publication year, follow-up period and study design) were tested to explore heterogeneity. This study considered a p-value <0.05 to be significant for the analysis. And then we conducted a subgroup analysis from the meta-regression result. 

12. Reporting biases are best performed only when you have a sufficient number of studies at your disposal. Please check the recommendations for such tests.

Remedy:

Yes, you are absolutely right and thank you for your valuable suggestions. It is true that reporting biases are best performed only when we have a sufficient number in this study. And insufficient number of studies was included in this meta-analysis. Hence, we deleted the publication bias and funnel plots in the manuscript. (Please see Line 340-343, Page 22)

13. The authors should provide a list with all included/excluded studies from their search together with the exclusion criteria.

Remedy:

Yes, thanks for your suggestion. We provided a list with all included/excluded studies from their search together with the exclusion criteria. (Please see the table S2)

14. Is it prudent to naively include/combine both RCTs and non-RCTs, since the latter can introduce additional bias in your results?

Remedy:

Thank you very much for the valuable consideration and comment. Admittedly, RCTs can provide the strongest and most epidemiologic evidence for causality if the RCTs are blinded[23]. In the present study, the non-randomized studies (NRS) were enrolled. It has been shown that the shortage of NRS may have strong likelihood of bias and confounding, data are more likely to be incomplete and of poor quality and outcomes are less likely to be validated[24,25] In particular circumstances, however, the non-randomized studies may provide certain advantages, such as providing us long-term information in early treatment of CIII malocclusion or maxillary transverse deficiency. Moreover, in ethical issue, the patients that are seeking for treatment due to their orthodontic problems and the observational studies may be more applicable in real-world settings than RCTs because of their broader range of participants included, large sample size and longer follow-up[23,26]. In the present analysis, nevertheless, there is few RCTs base available evidence. Alternatively, the studies we included were through the evaluation of risk of bias in individual studies (please see the Table 3).

15. What kind of cohort studies were included? Prospective or retrospective? The latter are usually more biased than the former.

Remedy:

Yes, thank you for your reminding. Of the 13 included cohort studies, nine [27-35]are prospective (including two partially prospective), and four [36-39] are retrospective, and all studies included untreated Class III controls. Two studies [28,35]are partially prospective, meaning that they present a retrospective control group (CG). We added which kinds of cohort studies into table 2. (Please see table 2)

16. Please provide additional details in Table I regarding the patients’ (how was skel. Class III defined), the treatment (appliance, duration, etc), and the outcomes assessed. Also relaying the actual conclusions of each study might be confusing, since they might disagree with the results of the meta-analysis.

Remedy:

Yes, thanks for your valuable suggestion. We re-organized the provide additional details in Table 2. In addition, in order to prevent the confusion when relaying the actual conclusions of each study, we decided to delete the authors’ conclusions. (Please see table 2) 

17. I am having issues with the risk of bias assessment presented. I checked randomly 2 included studies and I do not agree with the lenient assessment done with the authors. This needs to be checked to see if methods were appropriately applied.

Remedy:

Yes, thanks for your valuable suggestion. We re-checked and corrected the risk of bias assessment of the included studies in table 3. 

18. The authors need to provide in a table the full results of all meta-analyses performed, including studies, estimate with imprecision, p value, and heterogeneity statistics.

Remedy:

Yes, thanks for your valuable suggestion. We performed the summary results from primary and subgroup analyses as in table 4. 

19. How come the ‘explained variance’ is presented in table 3 but has not been described in the methods before? This is inappropriate. Also, reporting such a thing in meta-regressions that are not statistically significant is misleading. Please check with your statistician.

Remedy:

Yes, thank you for your valuable suggestions. It is true that it is inappropriate and misleading to reporting explained variance in meta-regressions. After checking with the statistician, we decided to remove this in the table. (Please see table 5, Line 294, Page 20)

20. I see no mention of assessing the quality of evidence with the GRADE framework, even though this is a standard approach for all systematic reviews nowadays.

Remedy:

Yes, thank you for your valuable suggestions. We have added the GRADE framework into this manuscript. (Please see Line 356-364, Page 22 and table 6)

21. I see no clear section outlining the limitation of the present study.

Remedy:

Yes, thank you for your excellent suggestions and through review. We have edited our manuscript and make clear section regarding the limitation of the present study. (Please see Line 424-431, Page 26)

22. Figure 2 is not necessary—you can give this information plainly in text.

Remedy:

Yes, thank you for your constructive suggestions. We followed your recommendations and gave the information plainly in text. (Please see Line 221 , Page 10)

 

Reference

1. Haynes S (1970) The prevalence of malocclusion in English children aged 11-12 years. Rep Congr Eur Orthod Soc: 89-98.

2. Thilander B, Myrberg N (1973) The prevalence of malocclusion in Swedish schoolchildren. Scand J Dent Res 81: 12-21.

3. Zhang W, Qu HC, Yu M, Zhang Y (2015) The Effects of Maxillary Protraction with or without Rapid Maxillary Expansion and Age Factors in Treating Class III Malocclusion: A Meta-Analysis. PLoS One 10: e0130096.

4. Arman A, Toygar TU, Abuhijleh E (2004) Profile changes associated with different orthopedic treatment approaches in Class III malocclusions. Angle Orthod 74: 733-740.

5. Guyer EC, Ellis EE, 3rd, McNamara JA, Jr., Behrents RG (1986) Components of class III malocclusion in juveniles and adolescents. Angle Orthod 56: 7-30.

6. Litton SF, Ackermann LV, Isaacson RJ, Shapiro BL (1970) A genetic study of Class 3 malocclusion. Am J Orthod 58: 565-577.

7. Ellis E, 3rd, McNamara JA, Jr. (1984) Components of adult Class III malocclusion. J Oral Maxillofac Surg 42: 295-305.

8. Liu Z, McGrath C, Hägg U (2009) The impact of malocclusion/orthodontic treatment need on the quality of life. A systematic review. Angle Orthod 79: 585-591.

9. Cunningham SJ, Hunt NP (2001) Quality of life and its importance in orthodontics. J Orthod 28: 152-158.

10. Almuzian M, McConnell E, Darendeliler MA, Alharbi F, Mohammed H (2018) The effectiveness of alternating rapid maxillary expansion and constriction combined with maxillary protraction in the treatment of patients with a class III malocclusion: a systematic review and meta-analysis. J Orthod 45: 250-259.

11. Bucci R, D'Anto V, Rongo R, Valletta R, Martina R, et al. (2016) Dental and skeletal effects of palatal expansion techniques: a systematic review of the current evidence from systematic reviews and meta-analyses. J Oral Rehabil 43: 543-564.

12. Cordasco G, Matarese G, Rustico L, Fastuca S, Caprioglio A, et al. (2014) Efficacy of orthopedic treatment with protraction facemask on skeletal Class III malocclusion: a systematic review and meta-analysis. Orthod Craniofac Res 17: 133-143.

13. Pithon MM, Santos NL, Santos CR, Baiao FC, Pinheiro MC, et al. (2016) Is alternate rapid maxillary expansion and constriction an effective protocol in the treatment of Class III malocclusion? A systematic review. Dental Press J Orthod 21: 34-42.

14. Rodriguez de Guzman-Barrera J, Saez Martinez C, Boronat-Catala M, Montiel-Company JM, Paredes-Gallardo V, et al. (2017) Effectiveness of interceptive treatment of class III malocclusions with skeletal anchorage: A systematic review and meta-analysis. PLoS One 12: e0173875.

15. Woon SC, Thiruvenkatachari B (2017) Early orthodontic treatment for Class III malocclusion: A systematic review and meta-analysis. Am J Orthod Dentofacial Orthop 151: 28-52.

16. Melsen B, Melsen F (1982) The postnatal development of the palatomaxillary region studied on human autopsy material. Am J Orthod 82: 329-342.

17. Franchi L, Baccetti T, McNamara JA (2004) Postpubertal assessment of treatment timing for maxillary expansion and protraction therapy followed by fixed appliances. Am J Orthod Dentofacial Orthop 126: 555-568.

18. Baccetti T, McGill JS, Franchi L, McNamara JA, Jr., Tollaro I (1998) Skeletal effects of early treatment of Class III malocclusion with maxillary expansion and face-mask therapy. Am J Orthod Dentofacial Orthop 113: 333-343.

19. Kim JH, Viana MA, Graber TM, Omerza FF, BeGole EA (1999) The effectiveness of protraction face mask therapy: a meta-analysis. Am J Orthod Dentofacial Orthop 115: 675-685.

20. Kapust AJ, Sinclair PM, Turley PK (1998) Cephalometric effects of face mask/expansion therapy in Class III children: a comparison of three age groups. Am J Orthod Dentofacial Orthop 113: 204-212.

21. Wolfe SM, Araujo E, Behrents RG, Buschang PH (2011) Craniofacial growth of Class III subjects six to sixteen years of age. Angle Orthod 81: 211-216.

22. Higgins JP, Thompson SG (2002) Quantifying heterogeneity in a meta-analysis. Stat Med 21: 1539-1558.

23. Kim HS, Lee S (2018) Real-world Evidence versus Randomized Controlled Trial: Clinical Research Based on Electronic Medical Records. 33: e213.

24. Sibbald B, Roland M (1998) Understanding controlled trials. Why are randomised controlled trials important? Bmj 316: 201.

25. Silverman SL (2009) From randomized controlled trials to observational studies. Am J Med 122: 114-120.

26. Edwards SJ, Lilford RJ, Braunholtz DA, Jackson JC, Hewison J, et al. (1998) Ethical issues in the design and conduct of randomised controlled trials. Health Technol Assess 2: i-vi, 1-132.

27. Chen XH, Xie XQ (2012) [The effect of two different methods of rapid maxillary expansion on treatment results of skeletal Class III malocclusion patients with maxillary protraction in early permanent dentition]. Shanghai Kou Qiang Yi Xue 21: 580-583.

28. Chong YH, Ive JC, Artun J (1996) Changes following the use of protraction headgear for early correction of Class III malocclusion. Angle Orthod 66: 351-362.

29. Kilicoglu H, Kirlic Y (1998) Profile changes in patients with class III malocclusions after Delaire mask therapy. Am J Orthod Dentofacial Orthop 113: 453-462.

30. Masucci C, Franchi L, Defraia E, Mucedero M, Cozza P, et al. (2011) Stability of rapid maxillary expansion and facemask therapy: a long-term controlled study. Am J Orthod Dentofacial Orthop 140: 493-500.

31. Masucci C, Franchi L, Giuntini V, Defraia E (2014) Short-term effects of a modified Alt-RAMEC protocol for early treatment of Class III malocclusion: a controlled study. Orthod Craniofac Res 17: 259-269.

32. Sar C, Arman-Ozcirpici A, Uckan S, Yazici AC (2011) Comparative evaluation of maxillary protraction with or without skeletal anchorage. Am J Orthod Dentofacial Orthop 139: 636-649.

33. Ucem TT, Ucuncu N, Yuksel S (2004) Comparison of double-plate appliance and facemask therapy in treating Class III malocclusions. Am J Orthod Dentofacial Orthop 126: 672-679.

34. Yuksel S, Ucem TT, Keykubat A (2001) Early and late facemask therapy. Eur J Orthod 23: 559-568.

35. Cozza P, Baccetti T, Mucedero M, Pavoni C, Franchi L (2010) Treatment and posttreatment effects of a facial mask combined with a bite-block appliance in Class III malocclusion. Am J Orthod Dentofacial Orthop 138: 300-310.

36. Akin M, Ucar FI, Chousein C, Sari Z (2015) Effects of chincup or facemask therapies on the orofacial airway and hyoid position in Class III subjects. J Orofac Orthop 76: 520-530.

37. Balos Tuncer B, Ulusoy C, Tuncer C, Turkoz C, Kale Varlik S (2015) Effects of reverse headgear on pharyngeal airway in patients with different vertical craniofacial features. Braz Oral Res 29.

38. Kajiyama K, Murakami T, Suzuki A (2004) Comparison of orthodontic and orthopedic effects of a modified maxillary protractor between deciduous and early mixed dentitions. Am J Orthod Dentofacial Orthop 126: 23-32.

39. Westwood PV, McNamara JA, Jr., Baccetti T, Franchi L, Sarver DM (2003) Long-term effects of Class III treatment with rapid maxillary expansion and facemask therapy followed by fixed appliances. Am J Orthod Dentofacial Orthop 123: 306-320.

---

## [Decision Letter · Decision Letter 1]

1 Feb 2021

Long-term maxillary anteroposterior changes following maxillary protraction with or without expansion: a meta-analysis and meta-regression

PONE-D-20-11899R1

Dear Dr. Huang,

We’re pleased to inform you that your manuscript has been judged scientifically suitable for publication and will be formally accepted for publication once it meets all outstanding technical requirements.

Kind regards,

Claudia Trindade Mattos, Ph.D.

Academic Editor

PLOS ONE

Additional Editor Comments (optional):

Reviewers' comments:

Reviewer's Responses to Questions

**Comments to the Author**

1. If the authors have adequately addressed your comments raised in a previous round of review and you feel that this manuscript is now acceptable for publication, you may indicate that here to bypass the “Comments to the Author” section, enter your conflict of interest statement in the “Confidential to Editor” section, and submit your "Accept" recommendation.

Reviewer #1: All comments have been addressed

2. Is the manuscript technically sound, and do the data support the conclusions?

Reviewer #1: Yes

3. Has the statistical analysis been performed appropriately and rigorously? 

Reviewer #1: Yes

4. Have the authors made all data underlying the findings in their manuscript fully available?

Reviewer #1: Yes

5. Is the manuscript presented in an intelligible fashion and written in standard English?

Reviewer #1: Yes

6. Review Comments to the Author

Reviewer #1: The authors accepted most of the suggestions and the article has improved substantially and is now suitable for publication.

7. PLOS authors have the option to publish the peer review history of their article (what does this mean?). If published, this will include your full peer review and any attached files.

Reviewer #1: No

---

## [Editor Report · Acceptance letter]

8 Feb 2021

PONE-D-20-11899R1 

Long-term maxillary anteroposterior changes following maxillary protraction with or without expansion: a meta-analysis and meta-regression 

Dear Dr. Huang:

I'm pleased to inform you that your manuscript has been deemed suitable for publication in PLOS ONE. Congratulations! Your manuscript is now with our production department. 

Kind regards, 

on behalf of

Dr. Claudia Trindade Mattos 

Academic Editor

PLOS ONE